# Epigenetic Mechanisms of Tree Responses to Climatic Changes

**DOI:** 10.3390/ijms232113412

**Published:** 2022-11-02

**Authors:** Barbara Kurpisz, Tomasz Andrzej Pawłowski

**Affiliations:** Institute of Dendrology, Polish Academy of Sciences, Parkowa 5, 62-035 Kórnik, Poland

**Keywords:** acclimatization, adaptation, chromatin modifications, climate change, DNA methylation, forest, histone modification, non-coding RNA, plasticity

## Abstract

Forest trees are complex perennial organisms that are adapted to the local environment in the results of prevailing climate conditions in population history. Because they lead a sedentary lifestyle, plants are exposed to various environmental stimuli, such as changes which can lead to the rapid adjustment or failure of their defence mechanisms. As forests play a key role in environment homeostasis and are the source of many products, it is crucial to estimate the role of forest trees’ plasticity mechanisms in the face of the climate change. Fast epigenetic adjustment is the basis for surviving climate fluctuations, however the question is whether this mechanism will be also efficient if climate fluctuations increase. Epigenetic modifications enable rapid reactions to the inducing stimulus by establishing chromatin patterns and manipulating gene expression without affecting the DNA itself. This work aimed to gather information about the epigenetic mechanisms of tree responses to changing environmental conditions, in order to summarise what is known so far and emphasize the significance of the discussed issue. Applying this knowledge in the future to study the interactions between climate change and gene regulation at the levels of plant development could generate answers to questions about the limitations of plasticity of plant adaptation to changing environment. We still know very little about how organisms, especially trees, cope with climate change and we believe that this overview will encourage researchers to fill this gap in the knowledge, and that results will be applied in improving defensive capacity of this ecologically and economically important species.

## 1. Introduction

Forest trees are distinguished from other plants by characteristic traits, such as longevity, relatively slow growth, long vegetative phases, and complex life cycles [1]. As these sedentary organisms are exposed to different stimuli, mechanisms of reactions to dynamic environmental changes are crucial. Temperature, water supply (particularly through precipitation), and the CO_2_ concentration in the atmosphere are key factors influencing the distribution and phenology of plants, including tree developmental processes as seed germination, flowering, and bud burst. These processes are associated with seasonality, and temporal shift in plant activity provides convincing evidence that species and ecosystems are affected by global environmental changes [2]. Synchronising the annual temperature cycle with the developmental cycle at the site of plant habitat is crucial for the survival of trees, particularly in cold and temperate regions [3]. Temperature increases and changes in rainfall may create particularly unfavourable conditions for the survival of some species, which may lead to a decrease in their current distribution [4]. The range of plants possessing phenotypic plasticity, due to having undergone adaptive evolution, ensures the maintaining of plant fitness across a range of environmental conditions. The mechanism underlying these acclimatisation responses can be driven by the epigenetic modifications [5]. Knowledge of these processes is important for understanding plant adaptation evolution and plasticity to changes in the habitat. Additionally, the understanding of the adaptation mechanism of trees to the various and changing environments and the ability to adjust the molecular, biochemical, and physiological processes to cope with them, can be helpful in the determination of the probability of survival of individuals, populations, and species.

Epigenetics is of primary importance for immobile plants, wherein changes in chromatin markers influence the expression of the most important genes regulating environmental responses [6,7,8]. Stress induces epigenetic changes in plants, enabling rapid adjustments in gene activity and expression patterns, which, in turn, lead to the plant’s ability to cope with the changes in its environment and reproduction [9]. Chromatin markers, such as DNA methylation [10], provide strong plasticity and modulate the development, morphology, and physiology of plants by constantly controlling gene expression and the mobility of the transposition elements [11,12,13]. Moreover, DNA methylation and histone modifications establish the functional status of chromatin domains and impart flexibility to transcriptional regulation, necessary for plant development and adequate responses to the environment [14,15]. Moreover, epigenetic changes may contribute to the ability of plants to colonize or persist in changing environmental conditions [16] and survive under stress by manipulating gene expression without affecting the DNA itself, i.e., the genetic information [17]. These conversions may occur spontaneously, be triggered by biotic/abiotic factors, or be due to the expression of other genes [18]. The stress caused by climatic changes can produce traces in the form of reversible, hereditary epigenetic changes that can be passed on to the next generation as a form of the maternal effect. This effect is known as epigenetic memory and is responsible for phenotypic variability and the plasticity of plants to new environmental conditions [19]. This memory may last for a long or short period, i.e., from a few days to several years, depending on the duration of the epigenetic change and the triggering stimulus; in some cases, it may extend to the offspring [20,21].

The epigenetic changes occurring in the F_0_ generation are inherited in the next stress-free F_1_ generation (intergenerational resistance) or in several subsequent generations (F_2_, F_3_, etc.; transgenerational inheritance) [20,22]. In the past, intergenerational inheritance has been defined as a form of ‘soft inheritance’ [23,24], as opposed to stable genetic inheritance induced by mutations. Epigenetic inheritance proves that in response to changing environmental conditions, a plant can make beneficial adjustments, which, in turn, can be passed on to the next generations. At the moment, when there are changes in the growing conditions, these changes are reversed. It is worth noting that the intergenerational epigenetic changes acquired by plants give them an evolutionary advantage through the survival of genotypes better adapted to adverse conditions [25].

Another well-described example of epigenetic memory in plants is priming, an adaptation strategy aimed at improving the plant’s ability to allocate metabolic energy to prepare its defence system for a faster and stronger response to future stresses. After triggering the priming phenomenon, the plant acquires a memory of stress that is epigenetic, resulting in a modification of the plant’s response to secondary stress or a permanent response to primary stress. The process of stress memory formation involves the storage of information about the inducing stressor through an epigenetic phenomenon and results in a modified reaction upon re-exposure to the stressor or a prolonged reaction after the induction of stress [20]. Priming consists of successive stages. In the first stage, plant receptors receive the priming stimulus, which may include abiotic stress, biotic stress, or chemical factors. Plant stimulation with an inducer causes a mobilization phase that induces changes at the physiological–biochemical, transcriptional, and epigenetic levels [26,27,28]. These changes lead to the plant storing information about previous stimulation in the form of a molecular record and they probably depend on many factors, including the intensity and duration of exposure to the stimulus, as well as the state and physiological condition of the plant [26,27]. Studying the priming phenomenon is problematic because the stress memory itself is revealed only at the last stage of priming, i.e., when a strong secondary stimulus is triggered. At this stage, changes occur in the plant that generate effective defence responses aimed at minimizing stress [29]. One of the advantages of this behaviour is that it generates low energy costs [28,30]. The hypothesis postulated by Bruce et al. [31] assumes that chromatin modifications occurring during plant sensitization to stress contribute to faster and increased accumulation of RNAs related to the priming phenomenon. However, if for an indefinite period the vegetation of plants proceeds in stress-free conditions, the changes that have occurred as a result of priming may be reversed [27]. This form of resetting the molecular record of previous stresses is, according to Crisp et al. [32], beneficial for maintaining the condition of plants at an optimal level.

This work aims to systematize and summarise the existing knowledge of the epigenetic mechanisms of adaptation and plasticity of trees to changes in climatic conditions. To date, not so many comprehensive studies of the processes related to climate relation of trees have been carried out based on epigenetic analysis. Three fundamental stages of tree development, controlled by environmental conditions, namely seed germination, bud burst and flowering, are reviewed in the context of the epigenetic control of climatic adaptation and plasticity. Additionally, the review presents needs and future perspectives concerning the investigation of the interactions between climate and gene activity regulation in plants and the importance of such knowledge in solving problems associated with environmental disturbances.

## 2. Plant Epigenetic Regulation Mechanism

The term ‘epigenetics’ was first used by Conrad Waddington in 1942 [33] to describe the interactions between genes and the environment, which, in consequence, was to lead to the development of certain phenotypes [30]. Considering the molecular aspect, the term ‘epigenetics’ means the study of heritable changes involving changes in gene expression, but not in DNA sequences [24,34]. Epigenetic changes are based on molecular processes, such as methylation of cytosine residues in DNA, chromatin remodelling by modifying histone proteins, and regulatory processes mediated by small non-coding RNA (Figure 1) [35,36].

DNA methylation is a stable modification consisting of the covalent attachment of a methyl group to the fifth position of the cytosine pyrimidine ring (5 mC) or the sixth position of the adenosine purine ring (6 mA). It is catalysed by DNA methyltransferase using S-adenosyl-L-methionine as a methyl donor [37,38,39]. This is important for various biological processes, mainly related to gene and transposon silencing. The number and location of methyl residues in the promoter or coding sequence of given genes have large functional consequences for this gene [38,40]. Modifications caused by DNA methylation in plants can be passed to the next generation through meiotic divisions or, reversibly, wiped out during mitotic cell division [41]. In plants, the methyl group (-CH3) is attached to the cytosine in three sequential contexts: (1) CG, catalysed by methyltransferase 1 (MET1) [42], (2) CHG (H = A/T/C), maintained by a plant-specific chromomethylase 3 (CMT3), in association with dimethylated lysine 9 in histone H3 (H3K9me2) [43] and (3) CHH, methylated by domain rearrangement methyltransferase (DRM2). DRM2-dependent DNA methylation is driven by small interfering RNAs (siRNAs) via the RNA-dependent DNA methylation (RdDM) pathway with consequent gene silencing [37,38,40]. Key factors related to these epigenetic modifications have been identified. There are Writers, which are enzymes responsible for the modification of nucleotide bases in DNA and amino acid residues in histone proteins [44]. The next group are Erasers, which are enzymes that erase changes established by Writers. The last group of factors are Readers, which are proteins with specific domains binding or interacting with epigenetic signs located in the locus [44].

Modifications of histones affect modifications of the chromatin structure in response to endogenous stimuli and changes in the environment [17]. Modifications of histone proteins include methylation, acetylation, phosphorylation, ubiquitination, and sumoylation. All of the above-mentioned modifications influence the formation of chromatin states [36]. One of the best-known histone modifications is methylation. Both lysine (K) and arginine (R) can undergo mono-, di-, or trimethylation (me1, me2, and me3) [45]. Histone methylation is mainly mediated by proteins containing the SET (Su(var)3–9, Enhancer-of-zeste, and Trithorax) domain. Modified histones are recognized by the corresponding proteins, which, together with other ATP-dependent remodelling complexes, make further changes in the availability of genetic information [46]. Highly conserved proteins from the Trithorax (TrxG) and Polycomb (PcG) families play a major role in regulating the expression of genes influencing developmental states in living organisms [47]. They also maintain the memory of the transcriptionally active or inactive chromatin status during stress [48]. TrxG and PcG play opposite roles in regulating gene expression of defence responses. It is generally accepted that the TrxG-mediated methylation markers associated with transcriptional activation, H3K4me2/3, maintain the trainable genes in a transcriptionally active state [49]. Thus, TrxG exhibits an antagonistic activity against the Polycomb family proteins which establish H3K27me3 and H3K9me2 methylation with a rather repressive effect on gene expression [50,51]. The chromatin structure may be regulated by histone methyltransferases and demethylases by the control of all degrees of lysine methylation, thus regulating various functions in the cell [52,53,54].

Histone acetylation was first described in 1964 by the Allfrey team [55]. The acetylation and deacetylation of the histone tails are mediated by enzymes known as acetylases (HATs) and histone deacetylases (HDACs), respectively. Many studies have found that HAT plays an important role in plant development and stress response [56]. Histone deacetylases (HDACs) are composed of three different families, of which HD2 are plant-specific deacetylases. Histone deacetylases lead to gene silencing through chromatin condensation and removal of the acetyl group from histone proteins [57]. Another modification of histones is phosphorylation. This modification is crucial for gene transcription activation, DNA repair, cell cycle-dependent chromosome condensation and segregation, as well as apoptosis [58,59]. Both serines (S) and threonines (T) are phosphorylated [60]. Another histone modification, ubiquitination, involves the attachment of a small conserved protein composed of 75 amino acids—ubiquitin to histone proteins. The linker histones and those that make up the nucleosomes are ubiquitinated [61]. Ubiquitin is attached to the ε-amino group of selected lysines by an isopeptide bond. This is a reversible phenomenon. Ubiquitination also serves as a signal for protein degradation that occurs in the proteasomes. It may also affect the subcellular localization and biochemical activity of the target protein [61,62]. Sumoylation is a modification of histones, in which the SUMO (small ubiquitin-related modifier) protein takes part. Like ubiquitin, the SUMO protein can covalently associate with other proteins using specific enzymes. Sumoylation is associated with the regulation of transcription and the stress response of plants [63,64].

Important in understanding the impact of chromatin modifications on gene regulation was the discovery that, even though different classes of epigenetic modifications act independently of each other, they can very often influence the recruitment of complex protein complexes regulating the transcriptional activity of genes in a complex manner [35,36]. The discovery of the importance of individual histone modifications led to the creation of a histone code that carries information about the activity of a given area of the genome. The histone code hypothesis assumes that a given modification of a non-specific histone residue determines modifications to the same or a different histone. Moreover, individual modifications or their combinations are “read” by protein complexes remodelling the chromatin structure, which influences the transcription of genes [65,66]. Chromatin remodelling complexes often use ATP to perform histone–DNA interactions, which, in turn, lead to changes in the nucleosomes. Proteins of the SWI/SNF class form the main complexes remodelling chromatin. The presence of DNA binding domains indicates the possibility of recruitment of these complexes by transcription factors [67]. In addition, these complexes play a key role in regulating growth and maintaining plant-specific dynamics of developmental changes. Remodelling complexes also interact with histone-modifying proteins, including from the TrxG and PcG family through bromo- or chromodomains [68]. In turn, the SWR complex is responsible for the exchange of histones in nucleosomes and the activation of gene transcription [60]. Regardless of whether the end effect is the activation or repression of transcription, this process is carried out through the complex action of complexes affecting histone proteins and transcription factors that act comprehensively [68].

RNA-directed DNA methylation (RdDM) is a biological process in which small, non-coding RNA molecules are involved in driving changes in DNA methylation through specialized transcription machinery [69,70]. RdDM depends on specialized transcription machinery clustered around two plant polymerases—Pol IV and Pol V [71]. In a process supervised by Pol IV, small interfering RNAs (siRNAs) are generated and transported to the cytoplasm. There they are attached to the AGO4 protein (ARGONAUTE 4) and re-transferred to the cell nucleus where siRNA directs the recruitment of Pol V transcripts and the targeting of DRM2 methyltransferase activity, consequently mediating de novo cytosine methylation in all sequence context classes (i.e., CG, CHG, and CHH; Figure 1) [69,70,72,73].

MicroRNAs (miRNAs) are small, endogenous, non-coding RNAs, about 20–25 nucleotides long, which are designed to suppress target gene expression through sequence complementarity [74]. miRNAs are involved in the inhibition of gene expression by directing the RNA-induced silencing complex (RISC) to mRNA on the basis of base complementarity [75]. By creating the RISC complex, the miRNA is attached to the AGO1 protein, consequently causing mRNA silencing through targeted cleavage or translation inhibition [76,77].

**Figure 1 ijms-23-13412-f001:**
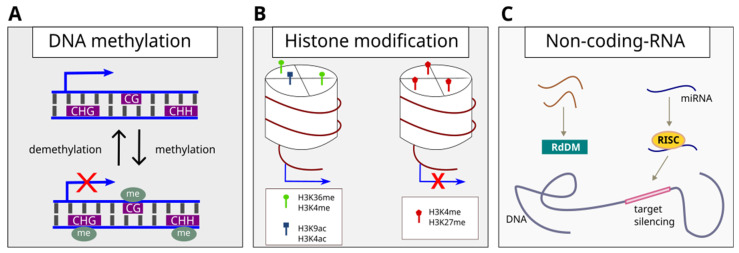
Epigenetic changes contributing to the plasticity characteristics in a plant to stressful environmental conditions (adapted from Villagómez-Aranda et al. [78] and Thiebaut et al. [79]). (**A**) DNA methylation. The process involves attaching a methyl group to the 5th carbon of cytosine in DNA. In plants, methylation occurs in three sequence contexts: CG, CHG, and CHH, where H = A/T/G. (**B**) Modifications of histones, involve post-translational modifications of the histone tails, mainly methylation and acetylation. It is assumed that on histone H3, methylation of lysine 4 and 36 and acetylation of lysine 4 and 9 is responsible for enhancing gene expression, while methylation of lysine 9 and 27 contributes to the inhibition of the expression of selected genes. (**C**) Non-coding RNAs. Small, about 20 nucleotide RNA molecules that do not code proteins, but regulate gene expression. They can participate in the RNA-directed DNA methylation (RdDM) pathway—small interfering RNA (siRNA), or by inducing a silencing complex (RISC) and binding to complementary genes on DNA, causing their silencing (miRNA).

## 3. Epigenetics in Tree Development

Forest trees are perennial organisms characterised by complex life cycles that are exposed to changing environmental conditions during their long lifespans [41]. From the end of the 20th century onward, Europe has been exposed to an increase in temperature combined with a deficit in rainfall, which has negatively impacted both the health and vitality of forest stands; this, in turn, may lead to significant social and economic losses [80,81] as forests play an essential role in maintaining the environmental balance, storing CO_2_, preventing soil erosion, supplying wood, etc. [82]. The adaptation strategies of forests to changing climatic conditions, including high temperatures and drought, is the paramount importance for the preservation of such properties [83,84].

Climate change is influencing the availability of resources and conditions that are critical to plant performance. Currently, some tree species are considered resistant to the effects of climate change [80]. One of the ways plants respond to changing environmental conditions is through the acclimatization by phenotypic plasticity [6,85]. Phenotypic plasticity is manifested through the ability of a single genotype to produce different phenotypes depending on the environment [86]. Recent studies provide important evidence that epigenetic mechanisms at the base of phenotypic plasticity are essential for stress responses (Table 1) [87,88,89] and can enable organisms to response rapidly to environmental changes, including climate change [90,91] (Figure 2). The loss or maintenance of epigenetic changes associated with changes in the environment of a particular plant enables a fair start for a new generation and ensures the growth and development of the offspring at the same level [92,93].

Silva et al. [94] investigating in silico DNA methyltransferases, DNA demethylases, and other histone modifiers in *Quercus suber* showed a link between the expression levels of each gene in different tissues (buds, flowers, acorns, embryos, cork, and roots) with the functions already known. They imply that the data generated during such investigation may be important for future studies exploring the role of epigenetic regulators in this economically important species.

**Table 1 ijms-23-13412-t001:** Selected epigenetic modifications of trees under the influence of environmental stresses.

Environmental Stress	Plant Species	Epigenetic Regulation	Plant Probes	Reference
temperature	*Picea abies*	DNA methylation; small non-coding RNAs (miRNA and ta-siRNA)	grafts; embryogenetic cultures	[13,95]
	*Populus tremula×alba*	DNA methylation; DML-dependent demethylation	adult trees	[96]
	*Castanea sativa*	gDNA methylation	3-year-old trees; shoots from trunk base (juvenile phase)	[97,98]
	*Malus × domestica*	DNA methylation	adult trees	[99]
	*Quercus suber* L.	DNA methylation/histone H3 acetylation	8-month-old plants	[100]
	*Populus tremula × tremuloides*	histone H3 acetylation	Tissue culture-grown plants transferred to soil and grown for six weeks	[101]
	*Picea abies* (L.) Karst.	epigenetic memory (DHNs)	adult trees	[102]
temperature, drought	*Populus deltoides × P. nigra*	DNA methylation	cuttings	[103,104,105]
temperature, photoperiod	*Picea abies*	DNA methylation	5-year-old plants	[106]
drought/water stress	*Quercus ilex*	DNA methylation	leaves from the upper part of the canopy	[107]
	*Populus tremula × P. alba*	RNAi suppression of DNA methylation	leaves/shoot apices	[108]
	*Fraxinus mandshurica* and *Fraxinus americana* hybrids	DNA methylation	2-year-old seedlings	[109]
	*Populus trichocarpa*	histone acetylation (H3K9ac)	3-month-old plants	[110]
salinity and nutrient variations	*Laguncularia racemosa*	DNA methylation	adult trees	[111]
the length of the day	*Picea abies* (L.) Karst.	miRNA	seedlings	[112]
radiation	*Pinus sylvestris*	genomic methylation, DNA methylation	plants from seeds	[113]

## 4. Dormancy and Germination of the Seeds

Research conducted by Reich and Oleksyn [114] indicated that the obtained from *Pinus sylvestris* trees had a specific memory of the climatic conditions in which their mother plants grew, which, in turn, affected their ability to germination in specific environmental conditions. Yakovlev et al. [91] provided evidence for the occurrence of an epigenetic memory phenomenon responsible for adaptation to variable environments in the common spruce (*Picea abies*). They indicated that the temperature fluctuations occurring during embryogenesis altered some characteristics of the adult individuals, such as frost tolerance, bud phenology, and seed production, suggesting a specific embryo-mature memory. Environmental factors during the storage of seeds in soil have a key impact on their later viability and germination which, in turn, can be passed to the next generation [115,116].

Alakärppä [117] conducted studies on DNA methylation and expression of selected DNA methyltransferase (DNMT) genes on mature *Pinus silvestris* L. seeds from three populations collected in northern and southern Finland. A correlation was found between climatic factors and the expression of DNMT genes in embryos, which may suggest that these changes contribute to the local adaptation of Scots pine. In addition, a variety of DNA methylation levels combined with changes in the expression of the studied genes may contribute to the improvement of the condition of trees in dynamically changing environmental conditions.

Seed dormancy is one of the most important elements of plant performance which delays the germination until optimal environmental conditions are appropriate for further growth and development. It is a complex trait caused by genetic factors and controlled by environmental conditions [118]. After seed dispersion, in soil seed banks, under natural conditions, the level of dormancy usually changes dynamically in the annual cycle, and the beginning of the growing season is associated with the highest seed germination potential. Climate changes (e.g., temperature and precipitation) may affect the durability of seeds stored in soil banks, targeting characteristics, such as their longevity, dormancy depth, and pathogen resistance [118]. The persistence of an exemplary seed population in a given environment depends on its resistance to premature emergence from the seed bank by germination or death and its exposure to the environmental conditions that favour this fate [119]. In this case, geographical distribution and rapid temperature changes may turn out to be unfavourable for a given species, given that the migratory capacity of woody plants affects their reaction in a limiting manner [118]. Changes in plant distribution ranges caused by climate change may not only result in migration to new areas that are more suitable for a given organism but may also select against phenotypes that adapt poorly to local conditions or disperse poorly [4,120,121]. Epigenetic regulations, which is the basis of plasticity, give the plants the ability to cope with the variability of the habitat conditions.

The seeds in the soil seed bank are constantly adjusting their dormancy to harmonise germination with climatic space and the season of the year [122]. In response to environmental stimuli, seeds show epigenetic changes that, in turn, result in the expression of dormancy-regulating genes. The team of Liu et al. [123] indicated that factors related to the PAF1C (RNA Polymerase II Associated Factor 1) complex, such as VIP4 (Vernalization Independence 4), VIP5, ELF7 (Early Flowering 7), ELF8, HUB1 (Histone Monoubiquitination 1), or RDO2 (Reduced Dormancy 2) are involved in the regulation of the dormancy in seeds and in early flowering. In addition, VIP4, VIP5, ELF7, and ELF8 are required for the expression of *FLC* (Flowering locus C), which can be regarded as a seed memory candidate gene due to its association with both flowering and seed dormancy. In turn, the HDA6 and HDA19 histone deacetylases are responsible for the regulation of germination by inhibiting the embryo-specific genes *LEC*1 (leafy cotyledon1), *FUS*3 (FUSCA3), and *ABI*3 (abscisic acid insensitive 3) [124,125]. Moreover, had2 and HD2A deacetylases correlate with the expression of the *ELO*3 (elongata 3) gene, which encodes a histone acetyltransferase in *Arabidopsis*, and the associated *DOG1* (delay of germination 1) gene [122]. HUB1 is a conserved ubiquitin-like protein and is required for the monoubiquitination of histone H2B at lysine 143 (H2BK143) [126,127]. It is a prerequisite for the trimethylation of lysine 4 (H3K4me3) and 79 (H3K79me3), which is related to gene activation [128]. Histone H2B monoubiquitination facilitates both transcription elongation and nucleosome refolding, and its loss leads to a reduction in the level of *DOG*1 transcripts in seeds, thus contributing to the subsiding of seed dormancy [129,130].

The persistence of seeds well adapted to changes in the ecosystem allows them to disperse over time and avoid the beginning of the germination phase until favourable conditions appear [131]. In the evolutionary context, delaying seed germination (bet-hedging strategy) spreads the risk of reproductive failure, which is especially important in an unpredictable environment where the risk of dying before reaching maturity is high [132]. A well-known genetic germination barrier is *DOG*1, a key dormancy regulator that determines the optimal temperature for seed germination [133]. *DOG*1 is specifically expressed in seeds and encodes a protein with unknown molecular functions. It belongs to a small family of proteins in *Arabidopsis* containing three conserved domains: PD87616, PD4114, and PD3883 [134,135]. Nakabayashi et al. [134] showed that DOG1 protein levels in mature seeds correlate with dormancy and remain stable during seed storage. *DOG*1 is alternatively spliced to produce four different cDNAs that are combinations of fragments of three exons. The functions of these isoforms remain unknown; however, their relative ratio does not change during seed development [135]. Cyrek et al. [136] showed that, as a result of alternative polyadenylation of the *DOG*1 gene, two mRNA variants of this gene are generated, short (sh*DOG*1) and long (lg*DOG*1). sh*DOG*1 in *Arabidopsis* is responsible for the production of the DOG1 protein and is, thus, responsible for establishing seed dormancy time [136]. Footit et al. [122] investigated changes in the chromatin of seeds from a soil seed bank and found that both the expression-activating sign H3K4me3 and the repressive sign H3K27me3 play a key role in temporal detection by regulating the expression of the *DOG*1 gene. Moreover, modifications of the histone H3 in the form of H3K4me3 and H3K27me3 established near the *DOG*1 gene are responsible for the thermal detection mechanism during the dormancy cycle. They found that trimethyl lysine 4 on histone H3 along the *DOG*1 gene is stable during dormancy maintenance (Figure 2) [122]. The repressive sign H3K27me3 slowly accumulates and accelerates upon exposure to light, ultimately leading to the loss of dormancy. Additionally, Müller et al. [137] focused on the observation of chromatin dynamics in key genes responsible for the regulation of seed dormancy, investigating two opposite signs of histone H3 methylation, i.e., H3K4me3 and H3K27me3. The mutual regulation of these signs was found through the transition from H3K4me3, responsible for the activation of gene expression, to the accumulation of repressive markers in the form of H3K27me3, which, in turn, persisted through the next stage of seedling growth. Thus, the transition to another phase of life is directly reflected in the change in chromatin levels, which is then sustained throughout further development [137].

ABI3 (abscisic acid-insensitive 3), the major regulator of the abscisic acid (ABA) signalling pathway, is a protein transcriptionally regulated at the chromatin level in *Arabidopsis* and in yellow cedar seeds. During the transition from dormancy to germination, the chromatin markers change from the active state (H3K4me3) to the repressive state (H3K27me3) [137,138,139].

*DOG*1 stimulates temperature-dependent dormancy, thereby influencing the levels of specific miRNAs [140]. Thus, *DOG1* can regulate dormancy by influencing the production and/or function and processing of the miRNAs miR156 and miR172, high levels of which inhibit (miR156) or promote (miR172) *Arabidopsis* seed germination at high temperatures (Figure 2) [140,141]. miR159c is involved in the control of MYB33 and MYB101 transcription factors, which positively regulate the ABA (abscisic acid) signalling pathway [142,143]. In addition, *DOG1* influences the expression of genes that code for miRNA processing proteins by inducing the transcript of the dicer-like1 (DCL1) enzyme and the hyponastic leaves1 (HYL1) RNA-binding protein and inhibiting the SERRATE (SE) protein [140]. Huo et al. [140] showed that the *DOG1* gene, involved in determining the seasonal germination time, influences also the flowering time of plants. Consequently, it provides a molecular mechanism that coordinates the response of dormant seeds and flowering plants with the environmental conditions.

## 5. Bud Dormancy

The regulation of the dormancy of vegetative buds is a complex process that is indispensable for the survival, development, and architecture of plants [144]. In tree phenology, the cessation of shoot growth is usually the first phenomenon indicating that the tree is dormant [145]. During the dormancy phase, the vegetative and reproductive meristems stop activity in order to withstand the harsh winter temperatures [145]. Rhode and Bhalerao [146] redefined the dormant state as a state in which cell division ceases and the meristem does not respond to growth-promoting stimuli until the plant is unable to induce growth from the meristems and other organs and cells, which can resume growth under favourable conditions. The production of vegetative buds by plants should provide adequate protection for regrowth or reproduction when present environmental conditions lead to the death of the actively growing or metabolising tissue. Plants have adapted as a result of evolution to precisely orchestrated signalling mechanisms that inhibit the growth and development of vegetative buds [146].

The dormancy of the buds is divided into three types: (i) paradormancy—the growth of the lateral bud is determined by the actively growing apical bud; (ii) endodormancy—determined by environmental conditions and endogenous factors; and (iii) ecodormancy—the plant is ready to grow but the prevailing adverse environmental conditions prevent active growth [147]. Ecodormancy begins in woody plants when the growth of the shoot apical meristem (SAM) and the cambial is stopped. The causes for the growth of these organs are low temperatures and short days. The next stage of bud dormancy is endodormancy, in which trees achieve maximum tolerance to low temperatures to survive the winter [96]. The epigenetic regulation of endodormancy suggests a possible role for chromatin conversion, which is similar to the epigenetic regulation of flowering after the vernalisation period [144]. The cracking of buds, which occurs during the time of awakening from the dormant state, depends on optimal temperatures, allowing the plant to grow and develop properly [148]. In a study conducted on poplars (*Populus*), the reactivation of SAM growth was preceded by a reduction in genomic DNA methylation in apical tissue, which, in turn, led to the induction of demeter-like 10 DNA demethylase (PtaDML10) in the apical buds of the dormant shoots. Furthermore, functional analysis showed that DML-dependent DNA demethylation mediated bud breakage [96]. Carneros et al. [102] demonstrated that memory based on epigenetic changes can convert the expression of bud crack genes in spruces, such as *EBB*1 (Early Bud Break 1) or *DHN* (dehydrins) and significantly influences the length of buds. Another example of a tree characterised by winter dormancy is the apple tree (*Malus* × *domestica* Borkh.), a fruit tree that grew in most temperate regions. In the winter season, low ambient temperature causes the tree to go dormant in order to survive the frost season. After sufficient cooling is obtained, the dormant buds wake up to resume normal active growth in high spring temperatures, and the transition from dormant bud to fruit set on the tree is accompanied by a decline in genomic DNA methylation. Lack of sufficient cooling in winter leads to irregular bud cracking and delayed flowering, which, in turn, has a negative impact on the yield and quality of the fruit [99].

The epigenetic control of bud break was described to be related to chromatin remodelling [144]. Inverse genomic DNA methylation and acetylated histone H4 patterns in inactive and active chestnut (*Castanea sativa*) buds provided information about the different forms of epigenetic control occurring during the transition through the different dormancy phases [97]. Decreased methylation was associated with bud growth initiation. Additionally, the level of acetylation of histone H4 (AcH4) was also higher during bud dormancy release (Figure 2). Both DNA methylation and histone H4 acetylation have been reported to play a role in the breaking of horse chestnut bud dormancy, and phosphorylation may also be involved in cell division that occurs after bud break in this species [97,149]. Analysis of the transcriptome showed that *HUB*2, encoding histone mono-ubiquitinase, and *GCN*5*L*, encoding histone acetyltransferase, were associated with bud dormancy, while *AUR3*, encoding histone H3 kinase, was associated with growth [149].

Fraga et al. [150] showed that an increase in the level of DNA methylation in *Pinus radiata* apical buds led to a state of progressive reinvigoration. A similar effect was observed during the development of the needles, i.e., the primary needles were characterised by a much lower level of DNA methylation compared to the mature needles. Younger tissues abound in signs associated with an increase in gene expression, e.g., trimethylation of lysine 4 on histone H3 (H3K4me3), while, in older tissues, they are replaced with repressive signs (e.g., H3K9me3) [151]. Additionally, in acacia (*Acacia mangium*), young buds were characterised, among other traits, by a higher degree of methylation in the micro shoots compared to other tissues. Epigenetic regulation was related not only to the shape or growth of an organ but also to the regulation of primary and secondary metabolism, i.e., photosynthesis [152,153]. Meijón et al. [154] studied the epigenetic modifications occurring in azalea (*Azalea japonica*) shoots and showed the opposition of DNA methylation and acetylation of histone H4, thus highlighting the specific dynamics occurring during the transition from vegetative to generative development. These studies allowed the delineation of the four basic phases occurring during the development of the azalea floral bud and the identification of the epigenetic reprogramming stage, characterised by a decrease in global DNA methylation.

Dormancy-associated genes DAM1, 4, 5, and 6 have been found down-regulated in flower buds of peach following dormancy release. These genes share a common chromatin modification involving H3K27me3 enrichment after dormancy release what shows a mechanism by which DAM genes might mediate growth and dormancy responses [155,156,157]. Lloret et al. [158] suggest that bud dormancy and stress tolerance share common regulatory epigenetic mechanisms, linking the dormancy stage with the environmental temperature.

## 6. Flowering

The onset of flowering is a critical life-history feature for a plant. Plants have evolved to flower at the time of year that provides them with optimal reproductive success in a given stand. Physiological studies have shown that flowering is initiated in response to both environmental signals linked to changes in temperature and day length and endogenous pathways linked to the developmental stage of the plant [159].

Plants growing in temperate climates experience a period of prolonged cold in winter before they can transit from vegetative growth and development to flowering in spring [160]. This process, referred to as vernalisation, allows for the induction of chromatin modifications in genes responsible for flowering, modifying the expression of these genes and, consequently, allowing the plant to flower in spring [84]. In the model plant *Arabidopsis*, vernalisation beneficially affects flowering by affecting the flowering locus C (FLC) epigenetic repressor of flowering involving PcG proteins. FLC encodes the MADS-box protein (Figure 2) [160,161,162,163]. The vernalisation process takes place in two stages. FLC repression occurs at low temperatures, to then be continued at 22 °C [163,164]. Low temperatures induce vernalization insensitive 3 (VIN3) activity, which is required for changes in histone modification and associated FLC repression [165]. Plant polycomb family proteins, i.e., fertilization-independent endosperm (FIE), vernalization 2 (VRN2), curly LEAF (CLF), and swinger (SWN), together with VIN3, form a complex that causes the trimethylation of lysine 27 of histone H3 in FLC in plants that have undergone vernalisation [162,165,166]. For accelerated flowering to occur in a plant, cell division must occur when low temperatures are acted upon [167,168]. Finnegan and Dennis [169] showed that the low-temperature treatment of plants inhibits FLC in mitotically inactive cells but this repression is not fully maintained. During cold treatment of plants, H3K27me3 is enriched at the beginning of the *FLC* gene before spreading at the locus after vernalisation, but this modification disappears after returning the temperature to 22 °C. This suggests the necessity of DNA replication to maintain the repression of the *FLC* gene.

The constans (CO)/flowering locus T (FT) regulatory module controls flowering time in response to changes in day length in annual plants. It also controls flowering in poplar [170]. Experiments by Böhlenius et al. [170] showed that the FT analogue in poplar, PtFT1, is a powerful inducer of flowering. The authors proved that the stems of young male aspen poplars (*Populus tremula × tremuloides*) transformed with *Agrobacterium* 35S::PtFT1 produced flower-like structures directly at the site of bacterial contamination within 4 weeks of transformation, while normal flowering times were in the range of 8 to 20 years.

## 7. Conclusions

Due to the dynamic changes taking place in our climate, it is important to understand the mechanisms of tree acclimatisation to environmental conditions. Factors, such as changes in temperature and precipitation, CO_2_ concentration, and human interference, significantly affect the phenological changes taking place in plants [2,118]. The synchronisation of the annual temperature cycle with the plant developmental cycle at sites of plant growth is crucial for the survival of trees, especially in cold and temperate regions [3]. One of the most important elements of plant adaptation to variations in climate is the phenomenon of plant dormancy. Delaying bud growth, seed germination, and flowering until optimal environmental conditions appear increases the chances of plant survival. Global warming has had a significant impact on these phases of plant reproduction and development, which, in turn, is disrupting the ecosystem [171]. Plants are sedentary organisms, so in order to survive, they adjust to environmental changes using epigenetic regulation, i.e., without direct interference in the genetic code. Stress-induced epigenetic changes can be passed on to the next generation. More and more researchers are focusing on delineating the relationship between the acclimatisation of trees to changing environmental conditions and epigenetic changes taking place in the plant. The juxtaposition of ecology, molecular biology, and epigenetics allows for an extension of the research perspective and an overview of plant processes at many levels. This approach may contribute to understanding the exact processes that occur in plants under stressful conditions caused by climate disturbances. Such a scientific basis may help determine the limits of tree plasticity in dealing with observed threats.

## Figures and Tables

**Figure 2 ijms-23-13412-f002:**
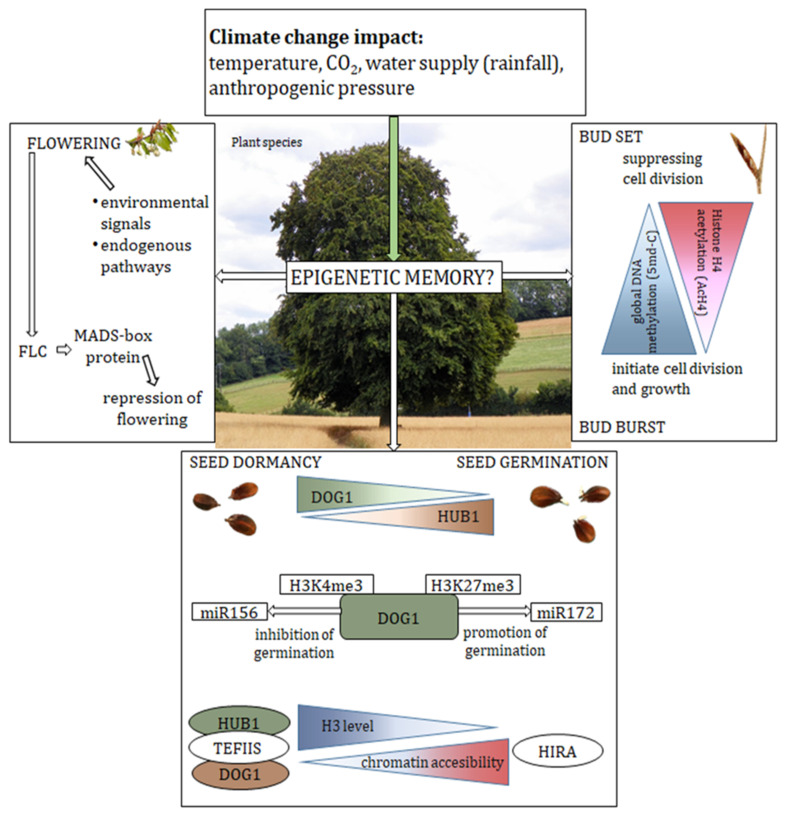
Epigenetic factors in the adaptive strategies of trees under the influence of environmental changes. Climate change caused by environmental factors such as temperature, CO_2_ concentration, water supply, etc., can influence epigenetic regulations in the plant. This, in turn, may result in the adjustment of the mechanisms responsible for flowering or the seed germination and bud burst. FLC, Flowering Locus C; DOG1, Delay of Germination 1; TEFIIS, Translational Elongation Factor 2; HUB1, Histone Mono-ubiquitination 1; HIRA, Histone Regulator A; H3K4me3, trimethylation of lysine 4 on histone H3; and H3K27me3, trimethylation of lysine 27 on histone H3.

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
