# Peer review of "Epigenetic Mechanisms of Tree Responses to Climatic Changes"

_ijms, 2022, doi:10.3390/ijms232113412_

Round 1

Reviewer 1 Report

The paper submitted by Kurpsiza and Pawłowski described the epigenetic regulation mechanism of forest tree adaptation to the environment from four aspects: tree development, bud and seed dormancy, and flowering, respectively. I think the topic selection of this paper is very important, and the description from the perspective of four important biological processes has certain novelty. The article cites a large number of important research findings, focusing on explaining the underlying methylation regulatory patterns associated with the formation of related traits.

The main flaw of the article are as follows, making the current article unable to meet the innovative requirements of IJMS,

1. it mainly focuses on the regulatory mode of DNA methylation, ignoring other epigenetic regulatory mechanisms;

2. it mainly focuses on the methylation levels of key regulators of trait formation, ignoring the synergistic effects of methylation and other regulatory pathways such as non-coding RNAs;

3. the research progress of forest trees is not sufficiently cited, and the epigenetic characteristics of forest trees are not fully revealed.

To overcome the above issues, it is recommended that the author make in-depth revisions to the article,

1. add a chapter to comprehensively describe the mechanism of plant epigenetic regulation;

2. from the aspects of tree development, bud and seed dormancy, flowering, etc., add description of the inheritance or reversal of epigenetic changes from F0 to the offsprings;

3. (use a list or other form) cite more the research progress of forest trees, or point out the direction or difficulty of the research;

4. figure 1 has no corresponding description in the text, and it is suggested to add more figures to convey more information.

Reviewer 2 Report

From my point of view, I suggest major revisions to improve the manuscript

Section introduction

1-     Write more details about how seed memory or priming can affect the plant development at different growth stages as well as the whole life cycle.

2-     Please write one paragraph about the previous studies related to candidate genes of seed memory in response to stimuli for different plant species.

3-     Please mention more details about the aim of this study.

4-     English grammar must be improved.

Reviewer 3 Report

The manuscript is a review about epigenetics in forest trees. the topic is very interesting. 

Explaining epigenetics need to carefully present what has been already shown experimentally, distinguishing annual model plants and forest trees, and presenting the actual limits of knwolege and gaps. The main risk is to over simplified epigenetics by overinterpreting few articles, especially reviews. Here, this mistake is totally present, and I gloablly disagree with most of conclusions that are over simplified.

In addition, the use of the literature is not correct for me, with some review articles used to demonstrate full concept or conclusions while even this reviews are more carefull. 

The structure of the paper and the choice are not clearly justified ... why some biological process in forest trees and not some others????

What is the input of this paper. I found 2 recent review papers on this topic of good quality.

I will encourage authors to rewrite entirely the review with a new original approach compared to the ones that are already published. 

here are just few examples of major questions :

Abstract is entirely to rewrite because gives evidence that are not so simple or so general on epigenetics and give a wrong vision of the topic. Give facts, hypothesis or challenge. The terms adaptation is never clearly used as genetic evolution or all process that accmimatize plants such as plasticity.

This sentence is too general and has no sense. All eucaryotes get epigenetics and whu woody plants will be the only with this « additional » adatative strategy

Many species of woody plants have an additional adaptive strategy for surviving climate change, namely the use of epigenetic modifications

Again too general. It is as if all is known about that and transgenerational epigenetics a classical phenomenon…. It is exagerated and fasle.

 These adaptations can either be passed on stably to the next generation or rapidly reversed. Despite the importance of this topic, currently, there is little published research on the epigenetic mechanisms of tree adaptation to changing environmental conditions

Why only these processes and not some others ? not clear about the choice of the focus

 at the levels of seed formation and germination, flowering, and bud development could

The problem is that it is not clear what is already available and not and what is the input of this review. Looking to recent reviews on the topic : Amaral et al., 2021 (cited here) and Garcia-garcia et al 2022 and some others (Sow et al., 2018…) i do not see the input here… at least it should be discusssed.

García-García, I.; Méndez-Cea, B.; Martín-Gálvez, D.; Seco, J.I.; Gallego, F.J.; Linares, J.C. Challenges and Perspectives in the Epigenetics of Climate Change-Induced Forests Decline. Frontiers in Plant Science 2022, 12, 797958

 Despite the importance of this topic, currently, there is little published research on the epigenetic mechanisms of tree adaptation to changing environmental conditions.

Introduction

This paragraph is symptomatic of the review. Epigenetic is oversimplified and using review papers (quite exclusively) or some experimental works, conclusions are made that do not reflect the reality of our actual understanding of plant epigenetics.

Indeed, epigenetics seems to be of pri-mary importance for plants, wherein changes in chromatin markers influence the expres-sion of the most important genes [9,10]. Stress induces epigenetic changes in plants, ena-bling rapid adjustments in gene activity and expression patterns, which, in turn, lead to the plant’s ability to adapt to changes in its environment and reproduction [11]. Chroma-tin markers such as DNA methylation [12] provide strong plasticity and modulate the development, morphology and physiology of plants by constantly controlling gene ex-pression and the mobility of the transposition elements [13–15].

Main text

Again, conclusions are simplified about epigenetics and plants…most of time from reviews… few years ago and no experimental recent data showing much more complexity if the situation.  The disctinction between what has been shown in arabidopsis and trees is not clear nd all is not possibl to transfert from one annual to perenials.

Epigenetic changes in the form of changes in chromatin structure can remain surprisingly stable across mitotic or meiotic divisions but can also be reversed relatively quickly [34,38].

I do not see the progression and relations among paragraphs… see for example section 2 : from Faga to Yakovlev … ???

Why plasticity is explained here but no littérature on stress in trees (drought…) and all the published works.  

Conclusion

You have to distinguish between plasticity and genetic evolution (adaptation)

it is important to understand the mechanisms of plant adaptation to environmental conditions.

Globally, one can expect to understand the actual limits of knowledge, the gaps, and the future challenges in trees concerning epigenetics…. But this needs to describe the real complexity of our actual understanding if epigenetics in plants and not over support all hypotheses.

Round 2

Reviewer 1 Report

I think the revision of the article are appropriate and acceptable for publication.